# Gait Compensation among Children with Non-Operative Legg–Calvé–Perthes Disease: A Systematic Review

**DOI:** 10.3390/healthcare12090895

**Published:** 2024-04-25

**Authors:** Abdulrhman Mashabi, Rula Abdallat, Mohammed S. Alghamdi, Mohammad Al-Amri

**Affiliations:** 1Department of Physical Therapy, College of Medical Rehabilitation Sciences, Taibah University, Al-Madinah Al-Munawarah 42353, Saudi Arabia; 2Department of Biomedical Engineering, Faculty of Engineering, The Hashemite University, P.O. Box 330127, Zarqa 13133, Jordan; rulag@hu.edu.jo; 3Department of Medical Rehabilitation Sciences, Faculty of Applied Medical Sciences, Umm Al-Qura University, Makkah 21955, Saudi Arabia; msghamdi@uqu.edu.sa; 4School of Healthcare Sciences, Cardiff University, Cardiff CF24 4AG, UK; al-amrim@cardiff.ac.uk

**Keywords:** Perthes, hip necrosis, gait compensation, gait adaptation, gait analysis

## Abstract

Perthes disease is a condition that affects walking patterns in young children due to poor blood circulation in the hip joint. Understanding the gait strategies of affected children is of great importance for an objective assessment and better management of this condition. The aim of this systematic review was to evaluate the current literature to identify gait compensation patterns in non-operative children with Perthes disease. Methods: A systematic electronic search was performed using MEDLINE, CINAHL, Embase, BIOSIS, and the Cochrane Library to identify studies published from inception up until December 2023. An adapted Downs and Black checklist was utilised to assess methodological quality and project risk of bias. Percentage agreement and nominal kappa statistics with bootstrapped bias-corrected 95% confidence intervals (CIs) were used. Result: A comprehensive literature search revealed 277 citations for review, of which 210 studies entered full-text screening. In total, eight studies met the inclusion criteria for quality assessment by two independent reviewers. The results revealed variations in data quality, with scores ranging from 12 to 17 due to missing information related to subject characteristics, biomechanical model, and power calculation. Conclusions: This review reveals common compensation strategies associated with walking among non-operative children with Perthes disease such as Trendelenburg gait due to weakness of the hip abductor muscle.

## 1. Introduction

Legg–Calvé–Perthes (Perthes) disease is defined as poor blood circulation in the hip joint due to idiopathic osteonecrosis. Patients with Perthes suffer mainly from pain associated with functional activities such as walking [1,2] and a limping pattern due to adductor muscle contracture or collapse in the epiphysis bone [2]. Several studies [3,4,5,6] have reported two distinct patterns among non-operative children with Perthes during single-limb support, Trendelenburg and Duchenne patterns. The “Trendelenburg pattern” is characterised by pelvic drop towards the non-affected swing limb, increased hip adduction, and trunk lean towards the affected stance limb; the “Duchenne pattern” is characterised by ipsilateral trunk lean towards the affected stance limb. Despite gait abnormality in children with Perthes being biomechanically considered in the previous literature, researchers debate the course of action required to manage trunk leaning. Westhoff et al. [4], Svehlik et al. [5], and Westhoff et al. [7] recommended performing ipsilateral trunk lean towards the involved side as an unloading mechanism to reduce the load in the involved hip joint; their recommendation was based on considering the hip joint only in their studies with no attention paid to the knee joint.

On the other hand, Stief et al. [6] considered knee adduction moment to evaluate the effect of ipsilateral trunk lean on knee joints among 27 children with Perthes and found that the effect of ipsilateral trunk lean was pronounced in the knee joint and could initiate degenerative changes in knee cartilage. They suggested that ipsilateral trunk lean should not be viewed as an unloading mechanism for the hip joint as it can cause excess lateral knee joint loading. Their findings may illustrate why children with Perthes suffer knee pain, as Nelitz et al. [2] reported. This conflict between recommendations may be due to a lack of studies investigating walking compensation mechanisms in patients with Perthes. To date, no published systematic review has been found discussing how non-operative children with Perthes might compensate for their walking. Investigating walking compensation mechanisms is essential to enable the clinical community to understand the underlying causes of gait deviation and decide on treatment plans to enhance walking. The objective of this systematic review was to identify the gait compensation strategies that have been described throughout the literature.

## 2. Methods

### 2.1. Search Strategy

This systematic review was carried out following the recommendations of the Preferred Reporting Items for Systematic Reviews and Meta-Analysis (PRISMA) guidelines [8]. A population, intervention, comparison, and outcome (PICO) approach was utilised. The population in this study comprised non-operative children with Perthes. The intervention was the motion analysis technique, the comparison was made with typically developing children (if found), and the outcome was gait parameters including temporospatial, kinematic, and kinetic in three planes (sagittal, frontal, and transverse). In order to provide a comprehensive overview of gait compensations, an electronic literature search was conducted within the databases MEDLINE, CINAHL, Embase, BIOSIS, and the Cochrane Library, using the search services Ovid, EbscoHost, Embase, and Web of Science from inception up until December 2023. This systematic review was registered in PROSPERO (CRD42018084815). The search strategy targeted the categories title, abstract, and keywords and includes the following search terms: gait, walking, locomotion, ambulation, mobility, compensation, adaptation, deviation, variation, alteration, changes, three-dimension motion analysis, two-dimensional movement analysis, gait analysis, kinematics, kinetics, spatiotemporal, angles, torques, moments, ground reaction forces, Perthes, hip necrosis, hip pathology, non-operative children, and non-surgical children (Appendix A). Wildcard symbols were used to retrieve all possible suffix variations of the root words. The search was restricted to the English language.

### 2.2. Selection Criteria

The search for literature was conducted in December 2023. The title and abstract of each study were screened, and full texts were retrieved subsequently and evaluated for definitive inclusion if they met the inclusion criteria (Table 1). Only Published full-length studies in peer-reviewed scientific journals from inception up until December 2023 were included, with the specific inclusion criteria determined a priori: (1) identified gait data; data had to be retrieved from skin-mounted markers by means of at least two-dimensional kinematic data; (2) walking on level ground or treadmills with a smooth surface and without any obstacles; (3) subjects walked freely without any kind of walking aid at either normal (self-selected), fast, slow, or default (e.g., paced) gait speed and either barefoot or in normal footwear (e.g., flat-heeled shoes); and (4) articles published in English. Reviews, conference papers, abstracts, letters, case series, and pilot studies were excluded. Since this systematic review does not aim to examine the effect of any intervention, observational studies (e.g., cohort, case–control, and cross-sectional design) were included in this systematic review. The inclusion criteria for the participants were non-operative children with Perthes aged between 5 to 12 years. Studies that presented individuals with any other musculoskeletal or neurological impairment and surgery in the lower limb were excluded. Two independent reviewers (R.A. and M.S.A., who are the co-authors of this article) screened the title and abstract of each study, and full texts were subsequently retrieved and evaluated for definitive inclusion if they met the inclusion criteria (Table 1). The grey literature, such as conference abstracts, non-peer-reviewed publications, and secondary literature, was excluded. Despite this grey literature being an important source of information for large-scale review syntheses, it would not be easy to search systematically, as Godin et al. [9] reported. Therefore, the grey literature was excluded from this systematic review study.

### 2.3. Data Extraction

All titles returned based on the search terms were first scanned by one of the co-authors, A.M. From the original search results, articles that did not meet the inclusion criteria were excluded (e.g., single video data without a marker, non-English language articles, including a running task, etc.). Following this, all titles and abstracts were reviewed independently by two reviewers R.A. and M.S.A. to determine their eligibility for the study. In case of disagreement between the two reviewers, the senior author (M.A.-A.) was consulted. The characteristics of the studies (authors and year), participants (sample size and age), gait parameters (temporospatial, kinematic, or kinetic), joints evaluated (thoracic, spinal, pelvis, hip, knee, or ankle), surface types (treadmill or overground), and gait compensation strategies were extracted and reported in Table 3.

### 2.4. Methodological Quality

The quality assessment (QA) of the included articles was performed based on the checklist introduced by Down and Black [10], which has been shown to have good inter-rater reliability (r = 0.75) as well as high internal consistency (KR-20:0.89). Since the included articles in the current study did not focus on treatment interventions, the checklist was adapted based on Schmid et al.’s [11] suggestion. The adapted Downs and Black checklist consisted of 17 items with a maximum score of 20 points, including the five different categories ‘‘quality of reporting’’ (eight items, maximum of ten points), ‘‘external validity’’ (three items, maximum of three points), ‘‘internal validity—bias’’ (three items, maximum of three points), ‘‘internal validity—confounding’’ (two items, maximum of two points) and ‘‘statistical power’’ (one item, maximum of two points). This adapted checklist was cross-validated by Schmid et al. [11]. Subsequently, the checklist was included in the data extraction sheet (Appendix B). All quality assessment data were extracted by R.A. and M.S.A. The reviewers received the relevant Perthes gait literature and the modified Downs and Black checklist by email in Excel format. Any disagreements were discussed to ensure consistency in the interpretation of scores. The extracted data presented in this systematic review include subject characteristics, methodological data, and gait compensation for children with non-operative Perthes disease in sagittal, frontal, and transverse planes.

### 2.5. Analysis

Percentage agreement and nominal kappa statistics with bootstrapped bias-corrected 95% confidence intervals (CIs) were used to ensure overall agreement between the two independent reviewers in the QA [12]. Kappa values were calculated using the command ‘‘crosstab” in IBM SPSS software Statistics for Windows, version 25 (IBM Corp., Armonk, NY, USA) [13]. Mean values along with standard deviations (SDs) were calculated for the summarised scores in each QA category to assess the included studies’ overall quality [11]. Meta-analysis of extracted data was not possible due to the heterogeneity of the gait outcome measurements.

## 3. Results

### 3.1. Selection of Studies

The electronic database search yielded a total of 277 papers. After removing duplicates, congress proceedings, non-peer-reviewed publications, secondary literature, and reviews, 210 studies were included for title and abstract screening. Following this step, thirteen full texts were retrieved and evaluated, of which eight articles met all the inclusion criteria (Figure 1).

### 3.2. Methodological Quality

The overall agreement between the reviewers performing QA on the literature in this review revealed a percentage agreement of 95% and a Kappa value of 0.906 (95% CI: 0.84–1), indicating “almost perfect” agreement based on Landis and Koch’s [14] study. The results of the modified Downs and Black checklist are presented in Table 2. The included studies are all based on 100% agreement between the two independent reviewers (R.A. and M.S.A.). In four studies (Westhoff et al. [4]; Yoo et al. [15]; Svehlik et al. [5]; Westhoff et al. [7]), items reporting gait analysis methods had to be rated as only ‘‘partially described’’ due to a lack of statistical information, patient characteristics, measurement device information regarding marker placement and device frequency, or the principal confounding factors (e.g., weight, height, and sex). Moreover, Yoo et al. [15] did not provide the actual probability value as the study did not include a control group. The identification of the source of the control population had to be rated ‘‘Unable to determine’’ in all of the studies. Another weakly scored item was the one reporting on staff, places, and facilities for the measurements, as none of the eight papers identified where the measurements took place or what the profession of the examiner was. Nevertheless, this item was scored as one (“Yes”) if the term laboratory was mentioned in the article or the affiliation. Finally, no study reported on the inclusion of a power analysis (a priori or post hoc) and the respective effective power values.

### 3.3. Methodological Data

All of the studies included in this review looked at walking strategies in patients with Perthes by the means of movement analysis systems to evaluate the walking pattern. Subject characteristics, methodological data, and gait compensation are presented in Table 3. The subjects age range was between 6 and 11.5 years, with an overall average group age of 8 years. Out of eight studies, six studies reported temporospatial parameters, which indicated that the patients walked more slowly than the control group subjects [3,4,5,6,16,17]. Only three studies reported subjects walking barefoot [4,7,16], while the rest provided no information about footwear [3,5,6,15,17]. Moreover, only three studies looked into both the affected and non-affected limbs of patients with Perthes [7,15,17], while the remaining five studies provided no clear information about the affected side. Overall, one study evaluated two joints [15], three studies evaluated three joints [3,4,17] and four studies evaluated four or more joints [5,6,7,16]. All studies considered kinematic and kinetic parameters, except Yoo et al.’s study [15], which only considered kinematic parameters. All studies considered the hip joint, whereby seven studies considered the pelvic joint, Stevens et al. [17] was the exception. Five studies looked into the trunk segment [3,4,6,7,16]. Four studies evaluated the ankle joint [5,6,15,17], and five studies considered the knee joint [5,6,7,16,17]. A summary of compensatory gait mechanisms in relation to the biomechanical constraints of the primary pathologies and the frequency of gait outcome measurement reported in Perthes gait literature are presented in Table 3 and Table 4.

## 4. Discussion

To our best knowledge, this is the first systematic review to assess the quality of Perthes gait literature and identify movement compensation strategies during walking among non-operative children with Perthes. The studies included in this review revealed consistent patterns of altered gait mechanics, including lower gait speed, shorter stride length, and longer stance phase. Notably, kinematic deviations are primarily observed at the pelvis and hip levels, with variations in trunk and pelvic obliquity, hip extension, and rotation. Some studies identified specific gait patterns, such as the Trendelenburg and Duchenne types, characterised by distinct pelvic and trunk movements. Kinetic analyses also indicate changes in hip and knee moments, with variations in positive and negative work, especially at the hip joint. The findings across these studies highlight the complexity of gait alterations in Perthes disease.

### 4.1. Biomechanical Gait Consideration

Several biomechanical gait considerations might have negatively influenced the results and the interpretation of the data throughout the included studies. The majority of the studies reported that children with Perthes walked significantly slower than the control group, except two studies that did not provide information on gait speed [7,15]. Yoo et al. [15] did not consider gait speed in their study, while Westhoff et al. [7] did not present gait speed data but reported no significant difference in gait speed between the Perthes group and controls; hence, the results do not indicate which of the two groups has the (non-significantly) slower walking speed. Several researchers encourage reporting gait speed in gait analysis research as speed changes gait patterns [18,19,20]. Therefore, to identify deviations in the kinematic and kinetic data, matching the gait speed of the control group subjects to that of the patient group is highly important to avoid misinterpretations of deviations solely due to gait speed. Schwartz et al. [21] investigated the effect of a wide variety of walking speeds on the gait of 83 typically developing children. They found that speed significantly influences temporal–spatial, kinematic, and kinetic parameters. On the other hand, it has to be taken into account that a reduced gait speed might already be considered a compensatory strategy. Therefore, gait speed should be reported in all gait studies as an essential outcome related to kinematic and kinetic parameters.

Another factor that is known to influence gait patterns is footwear. In this review, four studies [3,5,15,17] not provide any information on the footwear of the subjects. Murley et al. [22] and Radzimski et al. [23] conducted a systematic review to evaluate the effect of footwear on muscle activation in the lower limb and the influence of kinetic parameters. Murley et al. [22] evaluated 20 studies investigating the effect of footwear on lower limb muscle activity during walking by using an electromyography tool and found that footwear alters lower limb muscle activation. Radzimski et al. [23] evaluated 33 studies examining footwear modification as a conservative intervention to decrease the peak of external knee adduction moment and pain associated with knee pain. They concluded that footwear with lateral wedging was associated with the decreased peak of external knee adduction moment in healthy and osteoarthritic subjects. Moreover, this study found that subjects with footwear were more likely to increase the peak of external knee adduction moment than barefoot walking in healthy subjects. Therefore, barefoot walking is highly recommended for gait studies to eliminate the influence of footwear on gait analysis.

The often-missing evaluation of the unaffected side in patients with unilateral pathologies is considered a weak point of the reviewed papers. Given that the three studies that investigated both sides [7,15,17] found compensations on the unaffected side, the studies that evaluated only the affected side potentially missed compensatory mechanisms occurring in the other limb. Furthermore, evaluating more lower limb joints is also important to understand the nature of Perthes disease in walking parameters. The hip and pelvic joints were most considered in the Perthes walking literature, included in eight and seven studies respectively. The trunk and knee were considered in five studies, while the ankle joint was considered least in the Perthes walking literature, with it being included in only four studies. Based on the biomechanical of gait consideration section, future research is recommended to report gait speed, consider barefoot walking, evaluate the non-affected side, and evaluate more lower limb joints to better understand Perthes disease’s effect.

### 4.2. Identification and Interpretation of Compensatory Movement

#### 4.2.1. Temporospatial Outcome Measures

Perthes groups demonstrated slower gait speed [3,4,5,6,16,17], with short stride lengths [3,4,5,16,17] and prolonged stance phase time on the affected Perthes side compared to the controls [4,5]. This mechanism of lowering gait speed and prolonging standing time during the stance phase might be because the children feel unsafe; Hailer et al. [24] found that joint instability is responsible for a high incident rate of fracture and joint dislocation in Perthes subjects. Karimi and Esrafilian [25] also found that children with Perthes demonstrated postural instability due to hip muscle weakness. These findings suggest that gait stability might be affected in this group of patients. Van der Krogt et al. [26] investigated gait in 11 typically developing children and nine children with cerebral palsy under three different walking conditions: walking on a treadmill with a virtual environment, overground walking in a gait analysis lab, and walking in a natural environment outside the gait analysis lab; which established a link between reduced gait speed, short strides (length and width), and gait instability among children with cerebral palsy. As no studies in the Perthes literature consider the stride width parameter, gait stability is still unclear in this group of patients. Thus, it is important to further investigate gait stability and its link with falling risk and the prevention of injuries such as fractures [27,28].

#### 4.2.2. Kinematic Outcome Measures

The literature describes movement compensation with regard to three different planes, sagittal, frontal, and transverse, considering the trunk, spine, pelvis, hip, knee, and ankle. Different patterns were seen in the sagittal plane based on Perthes disease severity [5,7]. Children with Perthes demonstrated a flexion pattern in lower limb joints in the florid stage. Trunk movement showed a significant increase in total ROM. Relative to the pelvis, the trunk was in a significantly more pronounced posterior tilt position (“spine tilt”) compared to normal, while the movement and position of the trunk relative to the global coordinate system (“thorax tilt”) remained physiologic. At the pelvis level, maximum anterior tilt was significantly increased in both florid and advanced groups compared to normal. Minimum anterior tilt and total ROM were also increased, and there were significant differences at the hip joint level of both sides. On the involved side, total hip ROM was severely reduced compared to normal, which was related to reduced maximum hip extension; maximum hip flexion was normal. On the non-involved side, total hip ROM was increased compared to normal and the involved side due to an increase in maximum hip flexion. ROM was significantly reduced on the involved side at the knee joint level compared to normal due to reduced maximum knee flexion in the swing, but no significant deviations were observed on the non-involved side. Additionally, no significant differences were found at the ankle level, but the Perthes group demonstrated a reduction in ankle plantarflexion compared to the controls. In the advanced stage, there was no significant differences at the trunk, hip, knee, or ankle level compared to the controls and the non-involved side. This flexion compensation pattern has been reported as a protective mechanism of the affected joint to reduce joint pain. Frigo et al. [29] found that patients with juvenile chronic arthritis presented hip flexion with less knee extension, especially at the late stance phase, to reduce pain and protect the joint.

In the frontal plane, two distinct gait patterns were seen, both deviating from normal. The first is the Trendelenburg pattern, characterised by pelvic drop towards the swing limb and increased hip adduction with trunk lean to the stance limb in relation to the pelvis. The second is the Duchene pattern, characterised by trunk lean towards the affected stance limb, with pelvis lifting, hip abduction and external rotation in the stance phase of the gait. Trunk leaning is a notable sign of movement compensation among children with Perthes. This can be split into natural ipsilateral trunk lean (NTL) and excessive trunk lean (ETL). Thorax maximum obliquity was significantly increased in the Perthes groups, especially in the ETL group, compared to the control group. Pelvis maximum obliquity was decreased in the Perthes groups compared to the controls, but this difference was not significant. These deviations in the frontal plane may be due to hip and knee flexion during walking (as presented in the sagittal plane) and weakness in the hip abductor muscle. Krautwurst et al. [30] support the role of the abductor muscle in stabilising the pelvis in the frontal plane in cerebral palsy patients who demonstrated the Trendelenburg gait pattern. They found that hip abductor muscle weakness in children with cerebral palsy was associated with lower hip abductor moment and increased trunk lean to the ipsilateral side, while the pelvis remained stable as a compensation mechanism.

Furthermore, two distinct foot patterns were visible in the transverse plane, out-toeing and in-toeing. In all out-toeing patients, affected hips were externally rotated almost throughout the gait cycle, whereas the pelvis rotated internally. At the midstance phase, the external rotation of the affected hip increased compared with the unaffected side, and internal pelvic rotation also increased. At the midstance phase, internal rotation of the affected hip and external pelvic rotation increased compared with the unaffected side. In all in-toeing patients, affected hips persistently increased internal and external pelvic rotation during gait. Children with Perthes presented gait compensation in the transverse plane to avoid the femoral hump deviation impinging against the anterior acetabular rim during full flexion movement in the hip joint.

Another classification for children with Perthes was identified by Svehlik et al. [5]. They divided children with Perthes into three sub-groups (overloading, normloading, and unloading) based on hip joint loading. Hip joint loading was formed according to the extent of the time base integral of the hip abductor moments during the single-limb stance on the affected side in the frontal plane. First, the hip was in the extension position at the end of the stance in the overloading group and did not differ from the normal group. However, although mean hip adduction during single support was not different from that of the control group, its timing was abnormal, and the hip remained longer in adduction during the stance phase of gait. There was no rotational pathology of the hip in the overloading group. Pelvis motion was within normal limits in the sagittal and transversal planes. The pelvis dropped towards the swinging limb in the frontal plane and was not compensated for by the abductor muscles (as observed in normal gait). Second, the normloading group showed normal time-distance parameters except for prolonged stance duration. The hip did not reach normal extension at the end of the stance but was close to neutral in the frontal and transversal planes during single support. The pelvis revealed abnormal motion only in the frontal plane, where pelvis elevation on the swinging side was observed. Third, the unloading group walked with slight hip abduction during single support. Similar to the normloading group, the unloading group did not reach normal hip extension during walking. In the unloading group, in contrast to all other groups, the hip was externally rotated during single support, and internal rotation of the pelvis was documented. The elevation of the pelvis on the swinging-limb side was even more pronounced than in the normloading group.

#### 4.2.3. Kinetic Parameters

Svehlik et al. [5] stated that hip abductor moment is an important outcome measure to determine loading in the hip joint. As children with Perthes have poor lateral hip coverage, the peak contact pressure increases. In Westhoff et al.’s [4] study, the Trendelenburg group showed increased hip abductor moment compared to the control group, while the Duchene group demonstrated statistically significantly reduced abductor moment during the single-limb stance phase. In addition, Svehlik et al. [5] found that the overloading group demonstrated a higher hip abductor moment than the normloading group and the control group, while the unloading group displayed a reduced hip abductor moment during the stance phase. Trunk leaning may be another factor that influences hip joint loading. Stief et al. [6] and Stief et al. [16] found that hip adduction moment was lower in children with Perthes with ETL than those with NTL and controls, but these differences were not significant.

Another important kinetic factor is hip joint work. Westhoff et al. [7] and Stevens et al. [17] found that, in the florid stage, positive work in the hip joint was significantly lower on the involved side than on the non-involved side and in controls. Negative work was also reduced on the affected side compared to both the non-affected side and controls. This lower level of work done is an indicator of reduced activity of the hip muscles on the involved side and can be interpreted as an alleviation mechanism to reduce pain [7]. However, there is little work on knee and ankle joints. Only four out of the eight articles considered the knee joint, and one article considered the ankle joint. The kinetic parameters for knee joints revealed that knee adduction moment might be affected by trunk leaning pattern, as described in Stief et al. [6] and Stief et al. [16]. These works found that knee adduction moment was significantly lower in Perthes groups with trunk leaning, especially those with ETL, compared to controls. This altered gait pattern may increase the lever arm around the knee joint by shifting the GRF to the knee joint centre laterally. This lateral adjustment is argued to increase lateral tibiofemoral compartment load [31], which could be sufficient to deform the lateral compartment of the knee or influence the remaining growth plate and the physiological development of the mechanical axis of the leg in young patients and may lead to the development of knee osteoarthritis. Williams et al. [32] investigated late knee problems in myelomeningocele (spina bifida) patients. They assessed the incidence and aetiology of knee problems in a long-term follow-up of myelomeningocele patients. Seventeen patients with myelomeningocele out of 72 community ambulators had significant knee symptoms. They found that patients with myelomeningocele have a characteristic gait, which presents abnormal stress on the knee, leading to medial and anteromedial rotary instability and eventual degenerative change.

The kinetics of the ankle joint are given little attention in the literature on children with Perthes. Stevens et al. [17] is the only study considering the kinetics of the ankle joint. This study found that children with Perthes demonstrated significantly lower ankle work and power on both affected and non-affected sides than the controls; this was more pronounced on the affected side.

## 5. Clinical Implications

Children with Perthes demonstrated compensation strategies in all three anatomical planes and joints to alleviate pain, counter hip muscle weakness, and decrease the effect of the abnormal shape of the femur head. Effective treatment should focus on analysing these pathologic gait patterns and emphasise strengthening the hip abductor muscle to stabilise the pelvis and reduce hip joint stress. Studies have linked hip abductor weakness to increased adduction moments and compensatory gait changes. For instance, Horsak et al. [33] demonstrated that exercises strengthening hip abductors improved gait in children with obesity. Therefore, strengthening exercises are crucial for managing Perthes in children, potentially preventing complications like osteoarthritis and enhancing overall gait function.

## 6. Limitations

This review highlights a few limitations. The focus of this review was to assess the literature on gait analysis to identify gait compensation in children with Perthes. Within this context, we did not report the validated marker sets used by the included studies. However, this systematic review showed considerable variations in data quality, with inconsistencies in reporting subject characteristics, biomechanical models, and power calculations among the included studies. For instance, Yoo et al. [15] and Westhoff et al. [7] demonstrated gaps in providing comprehensive subject characteristics, control groups, and detailed information on biomechanical models and measurement devices. These issues may limit the reproducibility of the findings. Moreover, there was a lack of reporting information about power analysis, raising concerns about statistical power to detect meaningful effects based on this literature. The lack of standard reporting practices, as highlighted by Derrick et al. [34], further complicates the interpretation and comparison of results across studies. Future research should address these issues by adopting standardised biomechanical reporting guidelines and ensuring thorough documentation of subject characteristics, measurement methodologies, and control groups. Additionally, the varied classifications of Perthes disease and the limited available literature restrict the ability to compare across studies and conduct comprehensive meta-analyses. It is recommended that future studies include an evaluation of the unaffected side in patients with unilateral pathology to ensure a comprehensive understanding of secondary deviations.

## 7. Conclusions

Children with Perthes disease exhibit distinct compensatory movement strategies during walking, primarily aimed at alleviating pain, addressing hip muscle weakness, and mitigating the effects of abnormal femoral head shape. The identified compensatory mechanisms span three planes of movement—sagittal, frontal, and transverse—implicating various joints, with notable alterations observed in the hip and pelvic regions during single stance time. These adaptations result in slower gait speed, shorter stride lengths, and prolonged stance times, alongside increased hip adduction and pelvic drop towards the swing limb, which increase the load on the hip joint. Despite these compensations, hip joint loading remains elevated compared to controls, indicative of the complex interplay between structural abnormalities, muscle weakness, and adaptive gait patterns. The insights gained from this review underscore the importance of comprehensive motion analysis in identifying and understanding the multifaceted gait alterations in children with Perthes disease. Clinical interventions may recommend prioritising the strengthening of the hip abductor muscles to stabilise the pelvis, reduce hip abductor moments, and reduce compensatory movements. This approach aims to enhance gait function and prevent progression to secondary conditions such as osteoarthritis, thereby improving the overall quality of life for these children.

## Figures and Tables

**Figure 1 healthcare-12-00895-f001:**
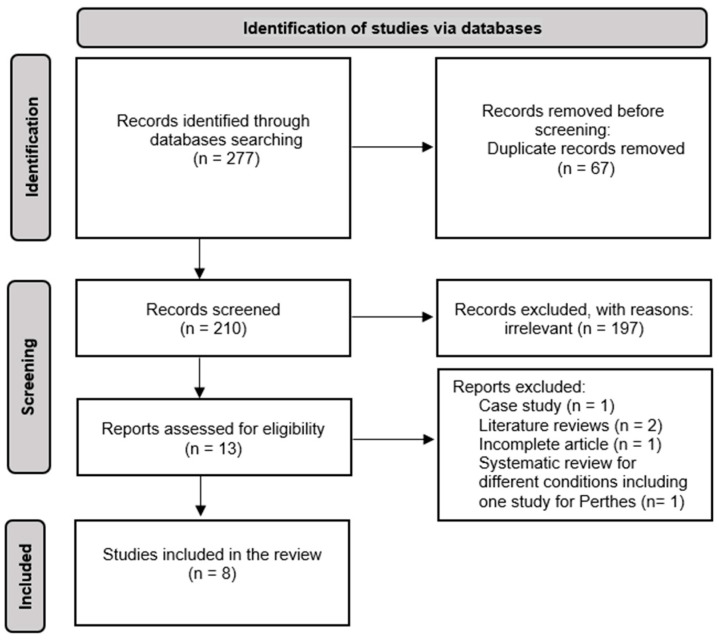
PRISMA Flowchart of the systematic search.

**Table 1 healthcare-12-00895-t001:** Inclusion and exclusion criteria for Perthes walking literature.

Category	Inclusion	Exclusion
Type of Study	Published in peer-reviewed scientific journals in the English language	Reviews, conference papers, abstracts, letters, cases series, and pilot studies
Main Outcomes	Clearly identified gait data; data had to be retrieved from skin-mounted markers by means of at least two-dimensional kinematic data	Single video data without markers
Subjects	Human cohorts presenting non-operative children with Perthes gait pattern aged between 5 to 12 years	Individuals with musculoskeletal impairment or surgery in the lower limb, individuals with a neurological impairment such as cerebral palsy or Down’s syndrome (due to complex cognitive impairments).
Measurement Conditions	Walking on level ground or treadmills with a smooth surface and without any obstacles	Stair climbing, walking uphill or downhill, and walking on uneven ground or a slippery surface
Walking Characteristics	Subjects walked freely without any kind of walking aid at either normal (self-selected), fast, slow, or default (e.g., paced) gait speed and either barefoot or in normal footwear (e.g., flat-heeled shoes)	Running studies with special footwear

**Table 2 healthcare-12-00895-t002:** Results of the study quality rated.

Author	Reporting		External Validity		Internal Validity (Bias)		Internal Validity (Confounding)		Power		Total Score	
	R.A.	M.S.A.	R.A.	M.S.A.	R.A.	M.S.A.	R.A.	M.S.A.	R.A.	M.S.A.	R.A.	M.S.A.
1. [4]	9	9	1	1	3	3	2	2	0	0	15	15
2. [15]	7	7	1	1	2	2	2	2	0	0	12	12
3. [5]	9	9	2	2	2	2	2	2	0	0	15	15
4. [7]	9	9	1	1	3	3	1	1	0	0	14	14
5. [6]	10	10	2	2	2	2	2	2	0	0	16	16
6. [16]	10	10	3	2	2	2	2	2	0	0	17	16
7. [17]	10	10	2	2	2	2	2	2	0	0	16	16
8. [3]	10	10	1	1	3	2	2	2	0	0	16	15
Total Mean (SD)	9.25 (1)		1.5 (0.6)		2.3 (0.4)		1.8 (0.3)		0		15 (1.4)	

**Table 3 healthcare-12-00895-t003:** Overview of gait compensation literature.

Study	Diagnosis	Number of Subjects (Gender) C: Control. P: Perthes	Age (Years) Mean (SD)	Parameters Evaluated	Joints Evaluated	Outcome Parameters	Perthes Mean (SD)		Control Mean (SD)	*p*-Value
1. [4]	A diagnosis of Perthes with unilateral involvement was confirmed on radiographs.	C: 30 (14 boys and 16 girls) P: 33 (24 boys and 9 girls)	C: 8.1 (1.2) P: 8.0 (2.0)	Temporospatial, kinematic and kinetic	Thoracic, spinal, pelvic, and hip	Gait speed (m/s) Stride length (m) Stance phase (s) At single-limb stance Thorax obliquity° Pelvic obliquity° Hip adduction° Hip abd moment (Nm/kg)	1.08 (0.19) 0.74 (0.07) 59 (2.2) Type 1–Type 2 −4.3 (5.7)–−5 (1.9) 4.3 (3.2)–0.2 (2.7) 9.4 (1.7)–1.1 (3.2) 0.43 (0.1)–0.24 (0.1)		1.18 (0.18) 0.8 (0.06) 58.3 (1.2) −0.8 (1) 1.5 (1.1) 4.9 (2.9) 0.40 (0.08)	*p* = 0.045 *p* = 0.001 *p* = 1.18
2. [15]	A 3D CT of the hip along with 3D gait analysis was performed to diagnose Perthes disease.	P, out-toeing: 5 P, in-toeing: 4 (7 boys and 2 girls)	P:11; range 7.0–15.3 years.	Kinematic	Pelvic and hip	At midstance phase Pelvic obliquity° Pelvis rotation° Hip flex/ext° Hip add/abd° Hip rotation°	Out-toeing Affected–Non-affected −3.3 (2.6)–−4.5 (4.4) 7 (5.6)–−13.8 (4.2) 10 (7.3)–7.8 (5.9) 2.5 (2.9)–3.5 (2.5) −10 (0)–3 (2.5)		In-toeing Affected_Non-affected −7.5 (3.5)–5.5 (1.9) −3.8 (2.9)–4 (3.9) 17.8 (6.8)–11.5 (6.2) 2.3 (3.6)–9 (±2.7) 12.5 (7.7)–1.5 (0.8)	
3. [5]	Diagnosis of Perthes disease was confirmed by X-ray.	C: 10	8.3	Temporospatial, kinematic, and kinetic	Pelvic, hip, Knee, and ankle	Gait speed (m/s) Stride length (m) Stride time (s) Cadence (steps/min) At single-limb stance Hip abduction ° Hip rotation ° Pelvic obliquity ° Hip abd moment (Nm/kg)	Overloading 1.65 (4) 0.93 (0.12) 1.02 (0.45) 130.1 (34.03) 5.96 (2.03) −3.54 (7.93) 2.39 (3.13) 22.5 (3.94)	Normloading 1.94 (3) 1.01 (0.16) 0.87 (0.07) 138.8 (12.67) 1.76 (3.6) −5.69 (8.21) −1.3 (3.63) 11.87 (2.23)	Unloading 2.07 (0.26) 1.05 (0.16) 0.85 (0.12) 144 (20.7) −1.8 (4.13) −12.8 (9.9) −2.22 (2.8) 3.72 (4.57)	Control 2.06 (0.3) 1.09 (0.11) 0.90 (0.12) 136.6 (15) 3.99 (2.35) −4.91(6.57) 1.5 (1.87) 12.6 (3.87)
		P: hip overloading 8 boys	11.4							
		P: hip normloading 19 (16 boys)	6.5							
		P: hip unloading 13 (11 boys)	7.6							
4. [7]	Perthes disease with unilateral involvement was diagnosed and confirmed by radiograph.	C: 30 (14 boys and 16 girls) P: 49 (38 boys and 11 girls) Group 1, florid: 36 Group 2, advanced: 13	C: 8.1 (1.2) P: 7.8 (2.3)	Kinematic and kinetic	Thoracic, spinal, pelvic, hip, knee, and ankle	Thorax ROM ° Pelvis ROM ° Hip ROM ° (florid) Knee ROM ° (florid) Hip positive work Hip negative work Knee positive work Knee negative work Ankle positive work Ankle negative work	Group 1–Group 2 4.4 (2.4)–3.4 (0.9) 6.2 (3.2)–3.5 (2.2) Affected–Non-affected 33.2 (9.2)–49 (6.7) 50.7 (7.1)–55.7 (5.6) 0.06 (0.03)–0.2 (0.09) 0.1 (0.06)–0.18 (0.1) 0.05 (0.05)–0.08 (0.06) 0.13 (0.07)–0.18 (0.10) 0.24 (0.09)–0.27 (0.09) 0.13 (0.05)–0.12 (0.04)		Control 3.2 (0.9) 2 (0.8) 44.5 (3.7) 55.7 (4.8) 0.15 (0.07) 0.15 (0.07) 0.06 (0.07) 0.18 (0.06) 0.28 (0.06) 0.13 (0.03)	Group1–control *p* = 0.018 *p* = 0.001 *p* = 0.001 *p* = 0.001 *p* = 0.001 *p* = 0.03 *p* = 0.636 *p* = 0.014 *p* = 0.052 *p*= 0.558
5. [6]	Children with ipsilateral trunk lean, including children with Perthes, were observed in two gait laboratories.	C: 20 (11 boys and 9 girls) P: 27 (19 boys and 8 girls) Group 1, natural ipsilateral trunk lean (NTL): 19 Group 2, excessive ipsilateral trunk lean (ETL): 8	C: 9.3 (2.3) P: 6.1 (1.8)	Temporospatial, kinematic, and kinetic	Thoracic, pelvic, hip, knee, and foot	Gait speed (m/s) Thorax max obliquity ° Pelvis max obliquity ° Foot rotation ° Hip add moment (Nm/kg) Knee add moment (Nm/kg)	NTL_ETL 0.48 (0.07)–0.44 (0.06) −4.3 (1.8)–−10.3 (3.5) 3.7 (3)–2.2 (3.4) −2.8 (10.7)–−7.3 (9.1) 0.60 (0.12)–0.51 (0.17) 0.29 (0.14)–0.16 (0.08)		0.49 (0.06) −3.2 (2.2) 5.9 (2) −7 (5.1) 0.73 (0.14) 0.47 (0.16)	*p* > 0.05 *p* < 0.001 *p* > 0.05 *p* > 0.05 *p* > 0.05 *p* < 0.05
6. [16]	Children with the unilateral diagnosis of Perthes confirmed on X-ray.	C: 19 (14 boys and 5 girls) P: 12 (10 boys and 2 girls)	C: 7 (2.5) P: 5.9 (2)	Temporospatial, kinematic, and kinetic	Thoracic, pelvic, hip, and knee	Gait speed (m/s) Step length (m) At midstance phase Thorax max obliquity ° Pelvis max obliquity ° Hip flex/ext ROM ° Knee flex/ext ROM ° Hip abd moment (Nm/kg) Knee abd moment (Nm/kg)	0.45 (0.05) 0.81 (0.06) −6.1 (3.2) 3.7 (3.3) 33.2 (9.8) 11.8 (4.1) 0.59 (0.18) 0.26 (0.18)		0.47 (0.06) 0.84 (0.06) −1.9 (2.2) 4.4 (2.9) 46.7 (6) 17.3 (6) 0.7 (0.13) 0.37 (0.15)	
7. [17]	A retrospective analysis of gait data including children with Perthes with other conditions.	C: 20 P: 45	C 22 (2) P: 14 (2)	Temporospatial kinematic, and kinetic	Hip, knee, and ankle	Gait speed (m/s) Step length (m) Ankle plantarflexion° Hip work (Nm/kg) Knee work (Nm/kg) Ankle work (Nm/kg)	Affected_Non-affected 0.27 (0.06) 0.36 (0.06)–0.56 (0.1) 13.8 (7.3)–21.6 (11.9) 0.31 (0.3)–0.56 (0.22) 0.14 (0.11)–0.22 (0.16) 0.66 (0.18)–0.69 (0.25)		Control 0.33 (0.03) 0.41 (0.03) 23.5 (8.7) 0.68 (0.18) 0.27 (0.1) 0.88 (0.26)	*p* < 0.05 *p* < 0.05 *p* < 0.05 *p* < 0.05 *p* < 0.05 *p* > 0.05
8. [3]	Children with unilateral Perthes disease classified by Mose classification based on latest follow-up X-ray.	C: 10 P: 10	C: 8.5 (2.3) P: 9.1 (2.1)	Temporospatial, kinematic, and kinetic	Thoracic, pelvic, and hip	Gait speed (m/min) Stride length (m) Cadence (steps/min) Thorax flex/ext ROM° Thorax add/abd ROM° Thorax rotation ROM° Pelvis flex/ext ROM° Pelvis add/abd ROM° Pelvis rotation ROM° Hip flex/ext ROM° Hip add/abd ROM° Hip rotation ROM° VGR force (N/BW)	57.4 (6.97) 1.06 (0.21) 107.6 (12.8) 11.12 (1.87) 14.04 (3.12) 16.85 (1.1) 10.26 (3.6) 8.25 (4.45) 18 (6.48) 40 (5.6) 13 (2.3) 14.7(12.2) 4.8 (1.7)		63.79 (8.1) 1.23 (0.15) 103.5 (7.7) 9.43 (3.52) 12.6 (3.82) 22.55 (3.33) 7.83 (3.12) 10.25 (4.2) 21 (10.46) 46.4 (5.6) 16.9 (9.3) 23.6 (8.8) 7.6 (2.5)	*p* < 0.05 *p* = 0.05 *p* > 0.05 *p* < 0.05 *p* < 0.05 *p* < 0.05 *p* < 0.05 *p* < 0.05 *p* < 0.05 *p* < 0.05 *p* < 0.05 *p* < 0.05 *p* < 0.05

C: control, P: Perthes, m: meter, s: second, min: minute, ROM: range of motion, °: degree, N: newton, Kg: kilogram, flex: flexion, ext: extension, add: adduction, abd: abduction, VGR: vertical ground reaction.

**Table 4 healthcare-12-00895-t004:** Summary of gait compensation.

Study	Gait Compensation
1. [4]	Temporospatial: Lower gait speed, shorter stride, and longer stance phase in the Perthes group. Kinematics: Two gait patterns; type 1 (trendelenburg) with pelvic drop, increased hip adduction, and trunk lean; type 2 (Duchenne) with trunk lean towards the affected limb. Kinetics: Type 1 showed increased hip abductor moment; type 2 showed reduced hip abductor moment.
2. [15]	Out-toeing: Marked decrease in hip internal rotation on the affected Perthes side. In all out-toeing patients, affected hips were externally rotated almost throughout the gait cycle, whereas the pelvis rotated internally. At the midstance phase, the external rotation of the affected hip increased in comparison with the unaffected side, and there was an increase in internal pelvic rotation. In the sagittal plane, flexion of the affected hips decreased during gait in all out-toeing patients; hip flexion at the initial heel contact decreased compared with the unaffected side. In the coronal plane, no gait deviation was observed in terms of hip adduction and pelvic obliquity. In-toeing group: Marked decrease in hip external rotation. The affected hips showed persistently increased internal rotation and external pelvic rotation during gait. At the midstance phase, internal rotation of the affected hips increased, and external pelvic rotation was compared with the unaffected side. In the sagittal plane, hip extension was decreased during gait in three patients: maximal hip extension decreased compared with the unaffected side. In the coronal plane, all affected hips in in-toeing patients showed increased downward pelvic obliquity.
3. [5]	Temporospatial: The normloading and overloading groups walked slower than the controls, while the unloading group walked faster. All Perthes groups had shorter stride lengths. Normloading and unloading groups showed less stride time, whereas the overloading group showed longer time. Both normloading and unloading groups had a higher cadence than the controls, with the overloading group having a lower cadence. Stance time was slightly longer in all Perthes groups. Kinematic: The overloading group demonstrated longer hip adduction during the stance phase, with normal pelvic motion except for a drop towards the swinging limb in the frontal plane. In the normloading group, the hip did not fully extend, with abnormal elevation of the pelvis on the swinging side. The unloading group showed slight hip abduction with external rotation during the stance phase, marked by more pronounced pelvic elevation.
4. [7]	Kinematics: The Perthes groups showed deviations mainly at the pelvis and hip level, which were more pronounced in the florid stage (Group 1). Group 1 showed a significant increase in trunk total ROM with a significant posterior tilt position compared to normal. At the pelvis level, both Perthes groups showed a significant maximum pelvis anterior tilt compared to the controls. At the hip joint level, Group 1 demonstrated a significant reduction in maximum hip extension on both affected and non-affected sides. ROM was significantly reduced on the involved side at the knee joint level compared to the controls due to reduced maximum flexion in the swing. On the non-involved side, there was no significant deviation. At the level of the ankle, no significant difference was found. In Group 2, there were no significant differences at the level of the trunk, hip, knee, or ankle compared to the controls and compared to the non-involved side. Kinetics: Positive work: In Group 1, total work carried out (mainly in the hip joint) was significantly lower on the involved side than on the non-involved side compared to the controls. Negative work: Negative work carried out in Group 1 (mainly in the hip and knee) was also reduced compared to both the non-involved side and the controls.
5. [6]	Temporospatial: Walking speed was slower in both NTL and ETL Perthes groups than the controls; however, there was no significant difference in walking speed between the Perthes groups and the control group. Kinematic: Thorax maximum obliquity was significantly higher in the ETL group than in the NTL and control groups. Pelvis maximum obliquity was lower in the ETL group than in the NTL and control groups. In foot rotation, less external foot rotation was seen in the NTL group than in the ETL and control groups. Kinetic: Hip adduction moment was lower in the ETL group than in the NTL and control groups. Knee adduction moment was significantly lower in the ETL group than in the NTL and control groups.
6. [16]	Temporospatial: Insignificant differences in walking speed and step length in the Perthes group compared to the controls. Kinematics: Higher trunk obliquity, reduced pelvis obliquity, decreased hip extension, and increase in maximum knee extension in the Perthes group. Kinetics: Decreased hip and knee and adduction moments in the Perthes group.
7. [17]	Temporospatial: Slower gait and shorter step length in the Perthes group. Step length was significantly shorter on the affected Perthes side and longer on the non-affected side than controls. Kinematic: Reduced ankle plantar flexion in the Perthes group. Kinetic: Reduced total positive work and lower hip and ankle work on the affected side.
8. [3]	Temporospatial: Slower gait and shorter stride length in the Perthes group. Kinematics: Significant differences in trunk and pelvis ROM and reduced hip ROM in all planes. Kinetics: Lower hip extension and adduction moment.

## Data Availability

No data is available for this study.

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
