# Peer review of "Gait Compensation among Children with Non-Operative Legg–Calvé–Perthes Disease: A Systematic Review"

_healthcare, 2024, doi:10.3390/healthcare12090895_

Round 1

Reviewer 1 Report

Comments and Suggestions for Authors

Dear Authors.

Congratulations. The subject of the study is highly relevant. However, the presentation of the manuscript must be strongly improved. In the abstract, the material and methods, and results must present more information. The Material and Methods must follow the guidelines PRISMA. The Discussion section must be shortened, and it is necessary to include a paragraph with the limitations of the study. There are more comments in the attached file.

Comments on the Quality of English Language

The English must be improved and there are corrections in some words. 

Author Response

Dear reviewer, 

Thank you for your valuable comments. Here, you can find my response. 

Reviewer's comment                               My response 

Add information about Methodological quality and risk of bias of the selected papers in the abstract.

Thank you for this suggestion. We have made this clear in the abstract.

I suggest rewritting the Material and  Methods using the Guidelines PRISMA in the method section.

We can assure the review that PRISMA was utilized throughout our research. Our apologize if this was not clear but we hope that the method section is now clearer in the revised version of the manuscript.

Add more information about the PRISMA and the strategy "PICO" in the search strategy.

Once again, thank you for this comment. The information about PRISMA and PICO have been add in the search strategy.

Improve the presentations of the references in the result table.

The presentations of the references in the result table have been improved.

Add a legend at the bottom of the Table to define all abbreviations indicated in the Table.

The legend has been added.

The Discussion is too long. I suggest reducing it. There are repetitions of many statements.

Thank you for this comment. We have reviewed our discussion we also noticed that there are a couple of parts that could be improved. We hope that the revised section has been improved with less unnecessary repetitions.

I suggest addding a paragraph with the limitations of the study at the end of the Discussion section.

Thanks for the comment. to make this clearer to readers we have added a separate limitation section.

Reviewer 2 Report

Comments and Suggestions for Authors

This manuscript is a systematic review conducted to identify gait compensation patterns in children with non-surgical Perthes' disease.

Please see the peer review comments on this manuscript below.

Title.

The term "Gait Compensation" is used in the title, but the main outcome in this manuscript is joint angle data during gait. I leave the decision to change the title to the author.

Abstract

"The results revealed large variations, and several methodological issues in these studies “. The authors should at least provide an overview of the methodological issues, since it is not possible for the reader to estimate what they are.

Discussion

If the study focuses on kinematic data, there should be a reference to the marker set.

Only in Table.4 the caption is embedded in the table and is difficult to find. It is recommended that the formatting be standardized.

Author Response

Dear Reviewer, 

Thank you for your valuable comments. Here, you can find my response. 

Reviewer's comment                                   My response 

The term "Gait Compensation" is used in the title, but the main outcome in this manuscript is joint angle data during gait. I leave the decision to change the title to the author

Thank you for the comment. In this systematic review we investigated all key gait parameters that include temporospatial, kinematic and kinetic parameters. We believe that any compensations in gait would include one or more than gait parameters and joint angles are one of the indictors of gait compensation. We feel that  gait compensation is the appropriate term on this occasion.

Abstract

"The results revealed large variations, and several methodological issues in these studies “. The authors should at least provide an overview of the methodological issues, since it is not possible for the reader to estimate what they are

Thank you for the comment. This has been rewritten to provide a clear information.

Discussion

If the study focuses on kinematic data, there should be a reference to the marker set.

Thank you for this comment. We have highlight this as a limitation of this study.

Only in Table.4 the caption is embedded in the table and is difficult to find. It is recommended that the formatting be standardized.

The legend has been added to the table 4.

Reviewer 3 Report

Comments and Suggestions for Authors

Dear editor, Thank you for your kind invitation to review the study entitled "Gait Compensation Among Children with Non-Operative Legg-Calve-Perthes Disease: A Systematic Review".

First of all, I would like to thank the authors for designing this study. 

The title and abstract of the study were written in accordance with the study design. 

I suggest them to use MeSH terms for keyword selection. 

In the introduction section, sufficient information about the study is presented.

In the method section, it is written in the search strategy section that the search was completed in 2022 and then updated again in 2023. Did the authors extend the date because they could not write the study? Please explain here.

The tables are very descriptive and clear. 

Did the researchers make any other statistical evaluation other than the statistical analysis presented in the studies, or was it only the analysis of the scores given by two authors?

In the Results section, the included articles are presented in detail. 

I think the discussion, clinical implications and research implications sections will be interesting and informative for the readers. It is very valuable for the authors to include their own clinical experience in the discussion.

In studies evaluating these gait parameters, it may be important to indicate whether the evaluations were performed before physiotherapy. Because with physiotherapy, muscle strength, flexibility and ROM may improve and gait parameters may change. 

I think that the study will be of interest to the readers. Taken as a whole, the article is written in a clear, understandable and fluent language.

Author Response

Dear Reviewer, 

Thank you for your valuable comments. Here, you can find my response. 

Reviewer's comment                                My response 

I suggest them to use MeSH terms for keyword selection.

Although MeSH terms were considered, the specific term for Perthes did not yield many relevant results in the MeSH database. As a result, to ensure a comprehensive and targeted search, alternative keywords were selected based on expert guidance from a librarian. This strategy was chosen to improve the thoroughness and relevance of the literature search, given the limited results encountered with MeSH terms for this particular topic.

In the method section, it is written in the search strategy section that the search was completed in 2022 and then updated again in 2023. Did the authors extend the date because they could not write the study? Please explain here

This has been rewritten to avoid confusion. The search for literature was conducted in December 2023

Did the researchers make any other statistical evaluation other than the statistical analysis presented in the studies, or was it only the analysis of the scores given by two authors?

It is just the analysis of the scores given by two authors.

In studies evaluating these gait parameters, it may be important to indicate whether the evaluations were performed before physiotherapy. Because with physiotherapy, muscle strength, flexibility and ROM may improve and gait parameters may change.

It is not mention in the included literature where the children with Perthes received physiotherapy or not. This is addressed partly in the limitation section.

Round 2

Reviewer 1 Report

Comments and Suggestions for Authors

Dear Authors. Congratulations. The presentation of the manuscript was improved. Only a small correction, change "weight" ...to ...."body mass" throughout the manuscript.

Reviewer 2 Report

Comments and Suggestions for Authors

I would like to express my gratitude to the authors for addressing the four comments I provided during the review process. Thank you for incorporating my feedback into the Abstract and Discussion sections. I have reviewed the changes made.